# CNN–RNN Network Integration for the Diagnosis of COVID-19 Using Chest X-ray and CT Images

**DOI:** 10.3390/s23031356

**Published:** 2023-01-25

**Authors:** Isoon Kanjanasurat, Kasi Tenghongsakul, Boonchana Purahong, Attasit Lasakul

**Affiliations:** 1College of Computing, Khon Kaen University, Khon Kaen 40002, Thailand; 2School of Engineering, King Mongkut’s Institute of Technology Ladkrabang, Bangkok 10520, Thailand

**Keywords:** COVID-19, pneumonia, chest X-ray, CT images, convolutional neural network, recurrent neural network

## Abstract

The 2019 coronavirus disease (COVID-19) has rapidly spread across the globe. It is crucial to identify positive cases as rapidly as humanely possible to provide appropriate treatment for patients and prevent the pandemic from spreading further. Both chest X-ray and computed tomography (CT) images are capable of accurately diagnosing COVID-19. To distinguish lung illnesses (i.e., COVID-19 and pneumonia) from normal cases using chest X-ray and CT images, we combined convolutional neural network (CNN) and recurrent neural network (RNN) models by replacing the fully connected layers of CNN with a version of RNN. In this framework, the attributes of CNNs were utilized to extract features and those of RNNs to calculate dependencies and classification base on extracted features. CNN models VGG19, ResNet152V2, and DenseNet121 were combined with long short-term memory (LSTM) and gated recurrent unit (GRU) RNN models, which are convenient to develop because these networks are all available as features on many platforms. The proposed method is evaluated using a large dataset totaling 16,210 X-ray and CT images (5252 COVID-19 images, 6154 pneumonia images, and 4804 normal images) were taken from several databases, which had various image sizes, brightness levels, and viewing angles. Their image quality was enhanced via normalization, gamma correction, and contrast-limited adaptive histogram equalization. The ResNet152V2 with GRU model achieved the best architecture with an accuracy of 93.37%, an F1 score of 93.54%, a precision of 93.73%, and a recall of 93.47%. From the experimental results, the proposed method is highly effective in distinguishing lung diseases. Furthermore, both CT and X-ray images can be used as input for classification, allowing for the rapid and easy detection of COVID-19.

## 1. Introduction

Toward the latter half of 2019, the coronavirus disease (COVID-19) started to spread across the globe. The World Health Organization declared COVID-19 a pandemic in March 2020 when the number of affected nations reached 114, and the number of positive cases and deaths reached more than 118,000 and 4000, respectively [1]. It demonstrates that pandemics are characterized by widespread diseases and high mortality rates. In the vast majority of pandemics involving infections caused by coronaviruses, real-time reverse transcription-polymerase chain reaction (RT-PCR) is one of the modalities used for diagnosis [2]; however, it does not have a high level of sensitivity and is available in a limited number of medical facilities. In addition, the delivery of the results may take anywhere between 24 and 48 h. Considering these factors, an alternative method for diagnosing COVID-19 is essential [3].

Deep learning (DL) is a method that has the potential to resolve a wide range of issues within numerous fields, as the associated models can take a variety of forms. For example, convolutional neural networks can serve as a base model for detecting intruders in substation power plants [4], while long short-term memory networks can be used for traffic flow forecasting [5] owing to their capacity to learn and remember long-term dependencies. In addition, recurrent neural networks are interconnected networks that form a direct circuit. Accordingly, the output of LSTM networks can function as the input to the active phase of the system. RNNs can process inputs of any length. Their computation incorporates historical data, and the size of the model does not increase as the length of the input increases [6]; all these features provide substantial advantages. Another sub-field of DL is the detection, diagnosis, and localization of lesions in medical images (e.g., radiography and magnetic resonance images), since DL is extremely effective in terms of computational time and yields a good diagnostic accuracy through the use of models that learn and make decisions based on simple data. Numerous studies, including those on the segmentation of brain tumors using DL [7] and COVID-19 radiographic enhancement techniques using CNN models [8], have demonstrated a high accuracy and rapid calculation ability of DL approaches.

Owing to the processing capability of DL and the need for chest X-ray or CT images, numerous researchers have studied the use of DL techniques for the diagnosis of COVID-19. In a previous study of COVID-19 diagnosis using X-ray images, DL was used on a small dataset. Using chest X-ray images, Zhang et al. [9] were able to identify COVID-19. They analyzed the data of 320 individuals diagnosed with pneumonia and 135 patients diagnosed with COVID-19. They achieved an accuracy of 91.24% using pre-trained versions of the VGG16 and ResNet50 models. Seven pre-trained models were utilized by Hemdan et al. [10] to diagnose COVID-19 from X-ray images. The images were used to compare the models’ performance. VGG19 and DenseNet201 fared the best, with 90% accuracy and 91% F1 score. Islam et al. [11] combined a CNN model with an LSTM network for categorizing lung disease (COVID-19, pnemonia, and normal) in X-ray images. The proposed model presented exceptionally good results, achieving an accuracy of 99.4% and a recall of 99.3%. However, they considered only using X-ray images, Rahman et al. [12] were able to identify COVID-19. They investigated the performance of a number of image enhancement techniques and a few different deep CNN models. The optimal method, which used gamma-corrected images and CheXNet, achieved an accuracy of 96.2% but the method can only classify images as COVID or non-COVID. Aslan [13] used two-step CNN methods to identify viral pneumonia, COVID-19, and normal cases from chest X-ray images. Initially, the DeepLabV3+ network used a chest X-ray dataset to semantically partition the lung sections in X-ray images, and image processing techniques, such as dilation, erosion, use of a Gaussian filter, and image thresholding, were used to improve the segmented output. The segmented lung images were then fed to the mAlexNet + SVM architecture, which divided them into three categories using mAlexNet for feature extraction and SVM for classification. This method achieved a classification accuracy of 99.8%. Although the method has a high classification efficiency, it required two different CNN networks to operate at its two stages: segmentation and classification, demonstrating that they use many process to provide output. In terms of CT techniques for COVID-19 lung images, Wu et al. [14] made an important contribution to the ResNet50 architecture with the multi-view fusion concept. In their investigation, 495 CT images were utilized. The authors devised a method that had a specificity of 61.5%, a sensitivity of 81.1%, and an accuracy of 76%. Xu et al. [15] were able to detect COVID-19 by utilizing ResNet18 with CT images; this method achieved a total score of 83.9%.

In previous studies, either COVID-19 X-ray or CT images were used. Therefore, a DL model suitable to a specific type of image must be selected. Perumal et al. [16] addressed the transfer learning from VGG19 and Haralick features to diagnose COVID-19 with the 205 X-ray and 202 CT images, which achieved 93% accuracy, 91% precision, and 90% recall on a small dataset. Hamed et al. [17] proposed a combined CNN–LSTM model with a multi-level feature extraction (MLFE) strategy involving GIST and scale-invariant feature transform (SIFT) to simplify the training of the CNN; this strategy aided in accurate COVID-19 detection and severity classification from CT and chest X-ray images. A total of 2390 CT images from a SARS-CoV-2 dataset were used for detecting COVID-19 (COVID or non-COVID) and 220 CT and X-ray images from an SIRM COVID-19 dataset for classifying the severity into four levels: mild, moderate, severe, and critical. A 98.94% accuracy in COVID-19 detection and 83.03% accuracy in severity classification were achieved. Their method has high performance in COVID-19 detection, but they evaluate it only on CT images.

Even though previous studies have presented diverse approaches to detect COVID-19 and classify lung disease, most of the research has considered only X-ray or CT images, not both. Moreover, several researchers evaluated their method using a small number of datasets, making it difficult to ensure that the performance would be replicated when these methods were tested on a larger dataset. Thus, this research aims to present the combined CNN-RNN network to distinguish three classes of lung disease (i.e., COVID-19, pneumonia, and normal) that can be used in both X-ray and CT images as input.

In this study, the CNN was combined with the RNN to improve the classification result because the CNN has high-efficiency characteristics for feature extraction but is unconnected across nodes within the same layer, while the RNN contains a property that analyzes dependencies and continuity from earlier information, which assists in pattern recognition from extracted features. The widely used CNN models VGG19, ResNet152V2, and DenseNet121 were combined with a version of RNN, including LSTM and GRU, to determine which combination of CNN and RNN architecture yielded the best results. There is no report that compares the performance of different CNNs when combined with different RNN versions. In this work, we collected a large dataset of 16,210 X-ray and CT images from various sources with varying image sizes, brightness levels, and viewing angles to use for experimentation, which guaranteed the method’s flexibility to handle input data and high reliability. Further, the image enhancement technique via normalization, gamma correction, and contrast-limited adaptive histogram equalization (CLAHE) was utilized to improve the original image quality. The approach was evaluated on the basis of its accuracy, precision, recall, and F1 score. Figure 1 shows a visual representation of the system’s overall architecture. Before and during enhancement of the image quality, X-ray and CT images of the patients’ lungs were obtained. This allowed us to better evaluate any potential issues with the patients’ respiratory system. The overall size of the images was reduced through pre-processing. Thereafter, the data were separated into two distinct groups: a training set, which was used to instruct the models, and a testing set, which was used to validate the accuracy of the models.

## 2. Materials and Methods

### 2.1. Data Sets

In this study, we utilized images from four widely published databases. From the first database [18], we used a total of 422 COVID-19 images out of 930 images sized 224 × 224 × 3, as only the front images were selected: 342 X-ray images and 80 CT images. From the second database [19], we used a total of 5140 (2545 normal images and 2595 COVID-19 images) out of 15,153 X-ray images sized 256 × 256 × 3 to balance the data. The third database was the CT scan database created by Kang [20]. We randomly selected 6859 CT images from 104,009 images sized 256 × 256 × 3, 512 × 512 × 3, and 1024 × 1024 × 3 to equalize the number of X-ray and CT images. These images were then separated into 2259 normal images, 2365 pneumonia images, and 2235 COVID-19 images. Thereafter, 3789 of 4273 pneumonia X-ray images from the database created by Kermany [21] with an image size between 400 × 138 × 3 and 2772 × 2098 × 3 were used. The remaining images were randomly excluded to balance the number of data in each class of lung diseases. In total, we used 16,210 images, which were divided into 9271 X-ray images (2545 normal images, 3789 pneumonia images, and 2937 COVID-19 images) and 6939 CT images (2259 normal images, 2365 pneumonia images, and 2315 COVID-19 images). Figure 2 shows example X-ray and CT images of COVID-19, pneumonia, and normal cases, while Table 1 displays the distribution of the X-ray and CT images within each classification.

### 2.2. Image Enhancement Techniques

#### 2.2.1. Normalization

Normalization improves an image, enlarging its brightness to fill the entire dynamic range to reduce the distribution of noise.

#### 2.2.2. Gamma Correction

Gamma correction adjusts the luminance intensity of an image with a non-linear transformation. This technique executes non-linear procedures on image pixels and reconditions the image saturation accordingly. It is crucial to maintain a constant gamma value.

#### 2.2.3. Contrast-Limited Adaptive Histogram Equalization

Contrast-Limited Adaptive Histogram Equalization [22] is an image processing method that improves images with a low contrast. Block size (BS) and clip limit (CL) are the two primary variables within CLAHE, which are primarily responsible for improvements in image quality. As input images typically have a very low intensity, increasing the CL causes the histogram of images to become flatter, making the images brighter. When the BS is increased, the dynamic range of images is stretched, increasing the image contrast. When image entropy is used [23], the two parameters obtained at the location of the maximum entropy curvature produce images with a quality regarded as subjectively favorable. Equalizing histograms for all contextual regions is one of the goals of CLAHE. The original histogram is trimmed, and the pixels that are cut off are redistributed to the various levels of gray. When compared with a standard histogram, a redistributed histogram stands out, as it limits the intensity of each pixel to a pre-determined maximum. An example of an enhanced image is shown in Figure 3.

### 2.3. Development of Combined Network

#### 2.3.1. Convolution Neural Network

Zisserman and Simonyan first proposed VGG19 [24]. This model includes a total of 19 layers, 16 of which are convolutional and 3 of which are fully connected [25]. A 3 × 3 convolutional kernel with a stride size of 1 pixel, 2 × 2 max-pooling to reduce the image size, and rectified linear unit (ReLU) to improve model classification and decrease computation time were used in this model; a 224 × 224 × 3 matrix was applied as the input.

ResNet152V2 is a version of ResNet [26]. It employs a skip connection and has 152 neural layers, allowing it to back-propagate and train deeper networks using the gradient. The two primary types of blocks in this network are identity blocks and convolutional blocks. ResNetV2 is distinguished from the original ResNet by the application of batch normalization to each weight layer before usage.

DenseNet121 is one of the dense convolutional networks proposed by Huang et al. [27] for the classification of images. It uses dense connections between layers through dense blocks, which connect all subsequent layers directly with the sizes of their feature maps for information transfer within the network. DenseNet121 consists of 121 layers, including a 7 × 7 layer, 58 3 × 3 layers, 61 1 × 1 convolutional layers, and a fully connected layer, with ImageNet-derived weights.

#### 2.3.2. Recurrent Neural Network

One type of an RNN is an LSTM network [28], which learns sequence order dependence. An LSTM network can differentiate between short- and long-term memories, store them, update or reveal them as needed, and solve the vanishing gradient problem. Input gates, output gates, and forget gates are all components of an LSTM cell. The values at specific intervals are stored in the cell’s memory. The data that can enter and exit the cell are restricted by the three gates. A GRU network has advantages over a regular RNN. According to Cho et al. [29], the reduced number of parameters of a GRU network makes it comparable to an LSTM network with a forget gate, as it lacks an output gate. A GRU network does not include distinct cell states, in contrast to an LSTM network. The streamlined organization of a GRU network facilitates training.

#### 2.3.3. Combined CNN-RNN Framework

The CNN models were placed first followed by the RNN models to distinguish COVID-19, pneumonia, and normal cases using both chest X-ray and CT images, as shown in Figure 4. Three CNN models (VGG19, ResNet152V2, and DenseNet121) were used to extract the important features. To reshape the CNN output to the RNN (LSTM and GRU) input, we reshaped the output of VGG19 (none, 7, 7, and 512), ResNet152V2 (none, 7, 7, and 2048), and DenseNet121 (none, 7, 7, and 1024) to (49, 512), (49, 2048), and (49, 1024), respectively [11]. In the fully connected layer, the dropout technique was used to avoid overfitting in the networks [30,31]. The final step was the application of the softmax function—the mathematical function used to calculate the probability of lung disease.

## 3. Experiments and Results

### 3.1. Data Pre-Processing

It was necessary to conduct pre-processing prior to training, as the images were obtained from multiple sources, and their sizes varied. Pre-processing is typically used to prepare input data to meet model requirements. The images were resized to 224 × 224 × 3 at the beginning of this stage, and data augmentation techniques, such as rotation, flipping, and skewing, were applied to increase the variety of available data and prevent overfitting during the training stage. After the conversion of the images to an array of pixels, the scale of each pixel was normalized to the interval [0,1].

### 3.2. Experimental Setup

In the experiment, each of the three pre-trained CNN models was combined with a version of the LSTM and GRU RNNs to distinguish lung diseases from the chest X-ray and CT images enhanced using normalization, gamma correction, and CLAHE. The datasets were separated into three sets, as shown in Table 1. In particular, 65% of the lung images in the dataset were used for training, 20% for testing, and 15% for validation. Both the training and testing phases used 32 batch size, and the datasets were trained on a combined network for 300 epochs using the Adam optimizer at 0.001 learning rate; all setting parameters were adapted from Appasami’s setup [32] that provide the best accuracy for training CNN to detect COVID-19 based on their experiment. All results were obtained utilizing the Keras (version 2.9.0) and TensorFlow (version 2.9.2) frameworks on Google Colab Pro (25 GB of RAM and a Tesla P100-PCIE-16 GB GPU).

### 3.3. Evaluation

We examined the accuracy, precision, recall, and F1 score to determine the performance of the proposed methods. True positives (TP) were defined as the number of images that were correctly classified; false positive (FP) as the value of negative but the model predicted it as positive; true negative (TN) as the number of images correctly identified as negative; and false negative (FN) as positive but the model predicted it as negative. Accuracy indicated the proportion of classification correctly identified and was calculated as follows: (1)Accuracy=(TP+TN)/(TP+TN+FP+FN).

Precision was defined as the measure of how often positive labels were assigned to the correct category and was calculated as follows: (2)Precision=TP/(TP+FP).

Recall (or sensitivity) was defined as the measurement of each correctly classified classification and was computed as follows: (3)Recall=TP/(TP+FN).

The F1 score was the average value of precision and recall and was computed as follows: (4)F1−score=(2TP)/(2TP+(FP+FN)).

### 3.4. Results

In this section, examples and comparisons of the performance of our proposed method for the detection of COVID-19, pneumonia, and normal cases using the pre-trained models are presented. These examples include the best approach based on the CNN models, such as ResNet152V2 with GRU in the original image, VGG19 with LSTM in the normalized image, and DenseNet121 with LSTM in the normalized image.

Table 2 presents the results of the tested methods, in which the overall performance of the best-performing classification approaches for COVID-19, pneumonia, and normal cases using the pre-trained models was compared. To ensure clarity, we reported the results only for the best-performing models.

Table 2 also details the optimal operating methods for the various pre-trained CNN models. The ResNet152V2 with GRU model utilizing the original images achieved the highest overall classification performance, with 93.37% accuracy, 93.73% precision, 93.44% recall, and 93.54% F1 score. It also achieved the greatest COVID-19 and normal classification efficiencies, with 94.14% accuracy, 90.58% precision, 94.64% recall, and 92.57% F1 score for COVID-19 cases and 93.65% accuracy, 92.49% precision, 87.36% recall, and 89.85% F1 score for normal cases. For the classification of pneumonia, the VGG19 with LSTM model using the normalized images achieved the highest accuracy, precision, and F1 score (99.35%, 99.26%, and 98.89%, respectively).

In terms of training time, the ResNet152V2 with GRU model utilizing the original images took the least amount of time to train the network—99 min and 57 s—followed by the DenseNet121 with LSTM model utilizing the normalized images and the VGG19 with LSTM model utilizing the normalized images—103 min and 55 s and 115 min and 1 s, respectively. Meanwhile, in terms of predicted time, DenseNet121 achieved the fastest time at 0.08 s per image, followed by VGG19 at 0.16 s per image and ResNet152V2 at 0.21 s per image. Further, the results of existing research studies on COVID-19 classification were compared with those obtained herein, as shown in Table 3.

Figure 5 depicts the overall suggested performance during the training and validation phases in terms of accuracy and loss. At epoch 300, the performance parameters of the ResNet152V2 with GRU model utilizing the original images were as follows: training accuracy, 94.92%; validation accuracy, 95.70%; training loss, 0.15%; and validation loss, 0.09%. Similarly, the VGG19 with LSTM model utilizing the normalized images achieved 96.09% accuracy during training, 95.31% accuracy during validation, 0.1 training loss, and 0.11 validation loss. The DenseNet121 with LSTM models utilizing the normalized images attained 91.8% training accuracy, 96.4% validation accuracy, 0.26 training loss, and 0.15 validation loss.

Figure 6 depicts the confusion matrix of our proposed architecture for the classification of COVID-19, pneumonia, and normal cases during the testing phase. ResNet152V2 incorrectly identified 215 images out of a total of 3242 images, including 67 COVID-19 images, 132 normal images, and 16 pneumonia images. The VGG19 with LSTM model utilizing the normalized images misread 250 images, including 103 COVID-19 images, 133 normal images, and 14 pneumonia images. The DenseNet121 with LSTM model utilizing the normalized images misdiagnosed 299 images, including 57 COVID-19 images, 234 normal images, and 8 pneumonia images. The ResNet152V2 with GRU model utilizing the original images yielded the best classification results for TP and TN.

## 4. Discussion

In this research, CNN models (VGG19, ResNet152V2, and DenseNet121) were cross-combined with RNN models (LSTM and GRU) for the detection and classification of COVID-19, pneumonia, and normal cases. According to previous research, Islam et al. [11] increased classification accuracy from CNN’s 99.0% to 99.4% by using a CNN–LSTM network. Hamed et al. [17] used a CNN-LSTM network to improve the classification accuracy of a GRU network from 94.63% to 98.94%. Yin et al. [39] found that the accuracy of ResNet20–RNN was higher than that of ResNet20 at 2.3%.

Another advantage of our method is flexibility, as it was able to classify between COVID-19, pneumonia, and normal cases from both X-ray and CT images. A comparison between proposed method and another methods in terms of accuracy, precision, recall, and F1-score is shown in Table 3. From Table 3 is demonstrated that the proposed method is more accurate than [14,15,33,34,35,36,37,38], which evaluated the effectiveness of their approaches only using X-ray or CT images. Perumal et al. [16] used X-ray and CT images for lung disease classification, but their methods were tested on a small dataset and provided an accuracy lower than the proposed method. However, Aslan et al. [13] reported a better method, since it not only had complex processes to deliver output but also used only chest X-ray images to classify COVID-19, pneumonia, and normal cases. Hamed et al. [17] achieved high performance, but they only discussed binary classification (COVID or non-COVID) based on CT images to classify, showing that their method is non-flexible for input format and has less categorical ability than the proposed method, which can classify three lung disease categories.

The limitations of our proposed method are the GPU utilization and high RAM consumption (minimum of 16 GB) for the model training. This is because our method involved a deep CNN network and combined it with an RNN model, which may not be suitable for low-resource devices.

## 5. Conclusions

In this study, we presented combined CNN–RNN networks for classifying COVID-19, pneumonia, and normal cases from X-ray and CT images, which are different types of diagnostic imaging. The image enhancement technique via normalization, gamma correction, and CLAHE was utilized to improve the original image quality. Each of the three pre-trained CNN models (ResNet152V2, VGG19, and DenseNet121) was combined with LSTM and GRU RNNs. To evaluate the proposed approaches, we collected a total of 16,210 chest X-ray and CT images from various sources, which had several sizes, brightness levels, noise, and angles of view, as shown in Figure 2. Approximately 65% of all image data were used for training, 20% for testing, and 15% for validation. In the analysis, the ResNet152V2 with GRU model using the original images performed the best, with 93.37% accuracy, 93.54% F1 score, 93.73% precision, and 93.44% recall. The ResNet152V2 with GRU model utilizing the original images also achieved the highest overall performance in terms of COVID-19 and normal case classification. However, in terms of pneumonia detection, the VGG19 with LSTM model utilizing the normalized images yielded the best results. Therefore, the utilization conditions should dictate which method is preferred. Finally, the proposed model can be used not only for the detection of COVID-19 but also for the analysis and diagnosis of diseases related to imaging.

In future work, we intend to improve the proposed method to diagnose more illnesses, such as lung cancer, and classify the severity of COVID-19 infection into asymptomatic, mild, severe, and critical.

## Figures and Tables

**Figure 1 sensors-23-01356-f001:**
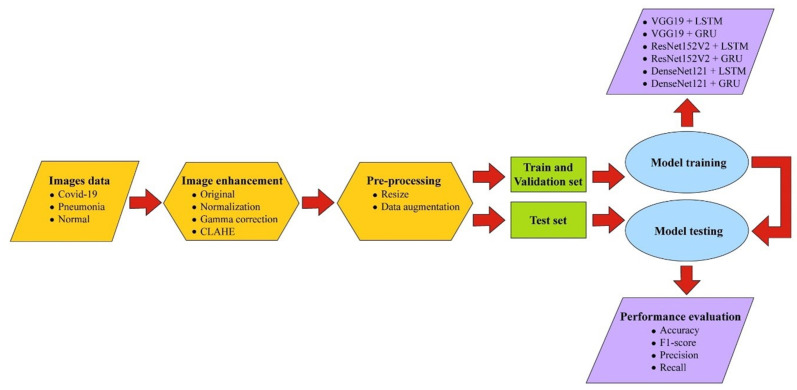
Block diagram of our research.

**Figure 2 sensors-23-01356-f002:**
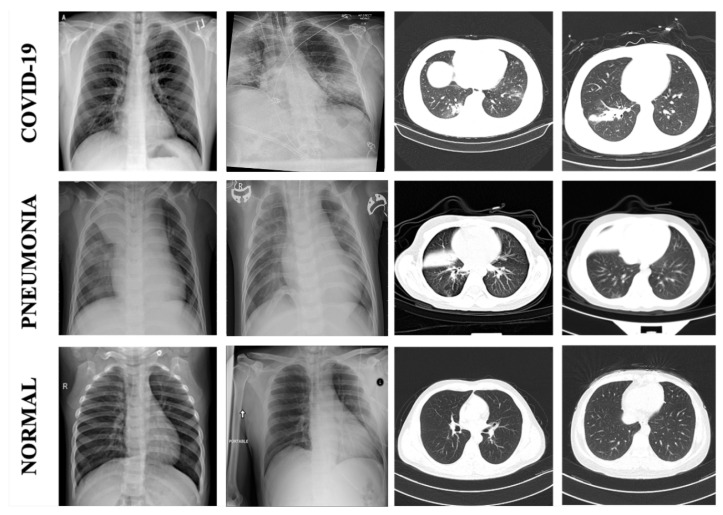
Example X-ray and CT images of COVID-19, Pneumonia, and Normal cases.

**Figure 3 sensors-23-01356-f003:**
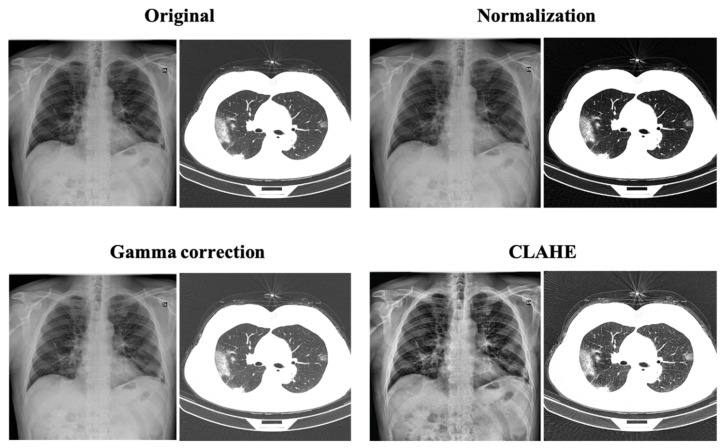
Original image vs. the images after various enhancement processes.

**Figure 4 sensors-23-01356-f004:**
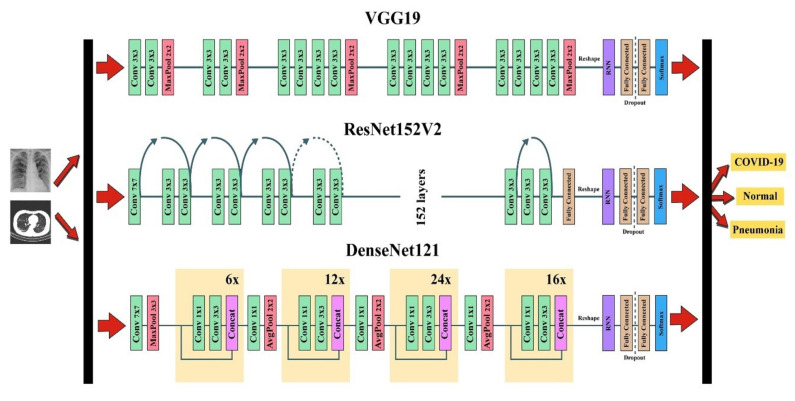
The combined CNN–RNN network structure.

**Figure 5 sensors-23-01356-f005:**
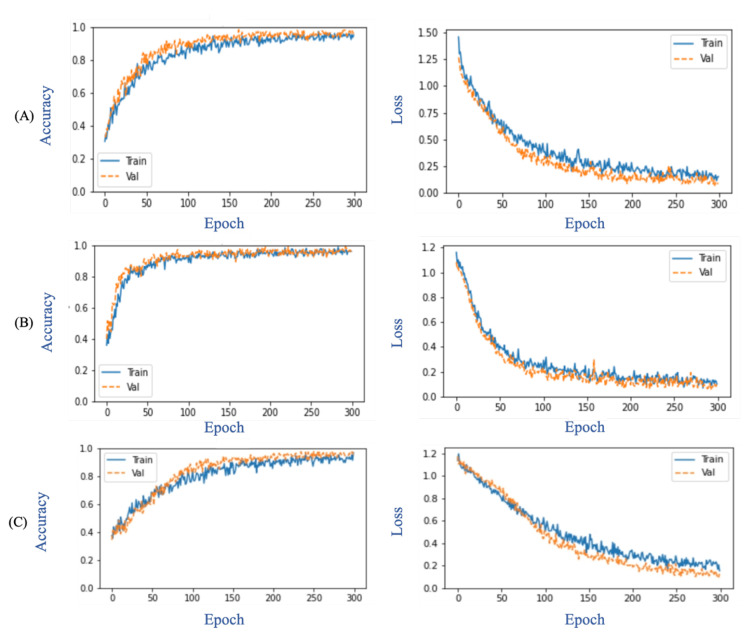
Evaluation metrics of CNN-RNN network: (**A**) ResNet152V2 with GRU in original image, (**B**) VGG19 with LSTM in normalized image technique and (**C**) DenseNet121 with LSTM in normalized image technique.

**Figure 6 sensors-23-01356-f006:**
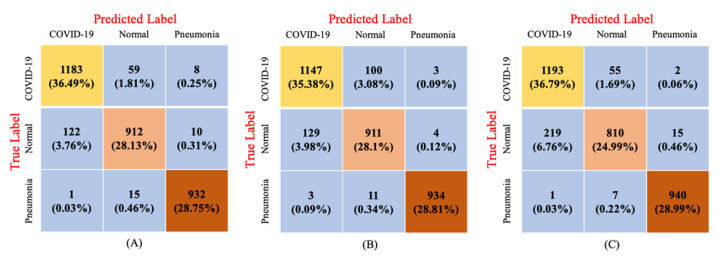
Confusion matrix of the best-performing methods: (**A**) ResNet152V2 with GRU in original image, (**B**) VGG19 with LSTM in normalized image technique and (**C**) DenseNet121 with LSTM in normalized image technique.

**Table 1 sensors-23-01356-t001:** The data set used in the experiment.

Data	X-ray			CT Scan			Overall
	COVID-19	Pneumonia	Normal	COVID-19	Pneumonia	Normal	
Training	1750	2713	1556	1452	1452	1452	10,375
Testing	750	398	600	500	550	444	3242
Validation	437	678	389	363	363	363	2593
Overall	2937	3789	2545	2315	2365	2259	16,210

**Table 2 sensors-23-01356-t002:** Comparison of CNN–RNN models for multi-classification network.

Model (CNN + RNN + Enhancement)	Patient Status	ACC (%)	Precision (%)	Recall (%)	F1-Score (%)	Training Times	Predict Times (/Image)
ResNet152V2 + GRU + Original	COVID-19 Pneumonia Normal Overall	94.14 98.95 93.65 93.37	90.58 98.11 92.49 93.73	94.64 98.31 87.36 93.44	92.57 98.21 89.85 93.54	99 m 57 s	0.21 s
VGG19 + LSTM + Normalization	COVID-19 Pneumonia Normal Overall	92.75 99.35 92.47 92.29	89.68 99.26 89.14 92.60	91.76 98.52 87.26 92.69	90.71 98.89 88.19 92.51	115 m 1 s	0.16 s
DenseNet121 + LSTM + Normalization	COVID-19 Pneumonia Normal Overall	91.45 99.23 90.86 90.77	84.43 98.22 92.89 91.85	95.44 99.16 77.59 90.73	89.60 98.69 84.55 90.95	103 m 55 s	0.08 s

**Table 3 sensors-23-01356-t003:** Comparison of results obtained in this study with other methods in the literature.

Author	Dataset Used (Class)	Method	ACC	Precision	Recall	F1-Score
Aslan et al. [13]	2905 X-rays (Multi-class)	Deep learning + SVM	**99.83**	**99.83**	**99.83**	**99.83**
Ozturk et al. [33]	625 X-rays (Multi-class)	DCNN	87.02	89.96	85.35	-
Asnaoui et al. [34]	6087 X-rays (Multi-class)	Inception+ ResNetV2	92.18	92.38	92.11	92.07
Rahimzadeh et al. [35]	15805 X-rays (Multiclass)	Xception + ResNet50V2	91.40	72.83	87.31	-
Saxena et al. [36]	13975 X-rays (Multiclass)	Modified CNN	92.63	95.76	91.87	93.78
Alshehri et al. [37]	746 CT (Binary)	Xception	84.00	-	91.70	-
Joshi et al. [38]	746 CT (Binary)	LiMS-Net	92.11	-	88.77	92.59
Wu et al. [14]	495 CT (Binary)	ResNet50	76	-	81.1	-
Hamed et al. [17]	2390 CT (Binary)	CNN-LSTM + MLFE	98.94	99.0	99.0	99.0
Xu et al. [15]	618 CT (Multi-class)	ResNet+ LocationAttention	86.7	81.3	86.7	83.9
Perumal et al. [16]	205 X-rays and 202 CT (Multi-class)	VGG16	93	91	90	-
**Proposed method**	**9271 X-rays and 6939 CT** **(Multi-class)**	**ResNet152V2+** **GRU**	93.37	93.72	93.44	93.54

## Data Availability

Not applicable.

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
