# Peer review of "CNN–RNN Network Integration for the Diagnosis of COVID-19 Using Chest X-ray and CT Images"

_sensors, 2023, doi:10.3390/s23031356_

Round 1

Reviewer 1 Report

Equation 1 maybee missing brackets if no fraction is used.

Both equation 1, 2 are too simple and well known, not neccessary to include in article.

Chapter 3.1 some error on the sentence end. ////////////////////////

Please describe exact version of used TensorFlow and Keras frameworks.

Hardware for training and learning was not mentioned, training time duration and inference delay missing.

Improove Abstract which is too much specific and Conclusion is very short without any of future works or improvements suggestions.

CTs and X-rays images could be divided to more group like beginning and advanced stage of the disease (usability in diagnostics is very questionable), but as initial research in this area can be accepted.

Author Response

Thank you very much for your comments and suggestions. Please see the response in the attachment.

Reviewer 2 Report

Using chest X-rays and CT images, Kanjanasurat describes CNN-RNN network integration for the diagnosis of COVID-19. It is very useful and can be improved for an interesting paper. However, currently, this work seems trivial, and there is no clear novelty in the work, the approach or the results. The concatenation model is quite simple, but there is no clear modification in the proposed model. Furthermore, the results of proposed model are not significantly higher than the previous research such as Perumal et al. [15] based on only VGG16 model although the number of datasets is lower than this work. Please add a critical discussion, why the proposed method provides higher performance than other research in terms of the networks in the model?  Prior to processing further, important issues must be addressed. It is necessary to revise points. It is interesting to consider if the authors provide detailed discussion and clarify the points after a major revision.

1. Please rewrite some complex sentences.

There are some grammatical errors: 

In abstract (line 11-12) Please add an article on precision and recall. : The ResNet152V2 with GRU model was calculated to have an accuracy of 93.37%, an F1-score of 93.54%, precision of 93.73%, and recall of 93.47%, respectively,

In abstract (line 14) Fragment sentence: Due to the method’s high level of dependability and comparable performance.

In introduction (line 19-21) Please add reference and rewrite the sentence.: 

The World Health Organization (WHO) addressed that it as a "public health emergency of international significance," and then, in March 2020, when the number of nations affected reached 114, there were more than 118,000 cases, and there were more than 4,000 deaths, the WHO classified it as a pandemic.

In introduction (line 23) Punctuation error: Pandemics are characterized by widespread disease and high mortality rates. [1].

In introduction (line 50) Punctuation error: on a small data set,Using images from chest X-rays, Zhang et al. [9] were able to identify 

In materials and methods (line 197 and line 202): There is no need to repeat "Accuracy (ACC)" because it is already mentioned in line 85.

2. For the abstract, add one sentence demonstrating a knowledge gap. Now that you've outlined what we already know, tell us what we don't know. How does your work address a specific question?

3. While it mentions that the method has "high level of dependability and comparable performance," it does not provide any information about what this means in terms of the accuracy, sensitivity, or specificity of the method.

4. To make the introduction and discussion more substantial, the authors may wish to introduce more examples and discussion. (such as 10.1016/j.asoc.2021.107918; 10.1016/j.chemolab.2022.104695, etc)

5. Please give some details about resolution each original dataset. And how to control.

6. Line 74-77: We increase the scope of disease diagnosis by making use of both X-ray images and CT scans. This allows us to take advantage of the computational speed and prediction accuracy of CNNs and RNNs, respectively, in order to forecast X-ray and CT scans of COVID-19 images.

7. Please add a result about "computational speed" ?

8. Line 105-106 "the remaining images were excluded, due to their small resolution." What is the minimum resolution required for this research? Please add references.

9. Line 193: Why this research use 300 epochs for training? Please add references.

10. Line 237: "During the testing phase, Figure 6 depicts the confusion matrix of our proposed". The "Figure 6" is not refer to the figure directly. Please edit it. 

11. Table 3. For Saxena et al. [31] show F1-score as 93.78 which is higher than the proposed method, so it should be bold. 

12. Figure 4. The combined CNN–RNN network structure. The figure shown the dropout dashed line, but there is no mention about "dropout" in the context. Please add more details and references.

13. Figure 4. The combined CNN–RNN network structure. Is the last layer of among VGG19, ResNet152V2, and DenseNet121 used same reshape parameters for combining with the RNN model? Please add more details and references.

14. For the Table 3, Is it fair to compare the other research with the number of dataset lower than the proposed method?

15. The authors should describe more clearly the ethics. For ethical considerations, experiments including human data and relevant approval protocols should properly be documented.

16. Please add limitation or drawbacks in this work?

17. For ‘Conclusions’, the challenges in studying this work, that the authors have addressed, should be emphasized.

Author Response

(The authors gave the same response as above.)

Round 2

Reviewer 2 Report

The authors have made an effort to address an important research question, and their work has the potential to contribute valuable insights to the field. However, the revised version of the manuscript has not addressed the key concerns raised in the previous report. As pointed out by the referee, the proposed approach lacks novelty and does not provide a clear explanation for its supposed superiority over previous papers. The lack of a comparison table that clearly summarizes the key aspects of the current work versus other reports is a major weakness of the manuscript. The authors MUST clearly state the novelty. What is unique, novel, or better about your work as compared to existing art? The table, which includes the summary and comparison of this current work versus other reports, should be added. (Table Columns: such as references, approaches, modes, data size, imaging techniques, etc.). The authors must claim: “there is no report on…. Now, no clear sentences in the abstract and introduction showing the novelty and the gap of knowledge that still requires you to just simply expand the scope of disease diagnosis by using both X-ray and CT images. Showing only a minor improvement (Table 3) is not enough to show the motivation and significance to be a new publication. A submission may be rejected if it is not significant to the readership, even if it is scientifically sound.

Furthermore, the authors have failed to provide a detailed letter addressing the specific points raised by the referee. It is essential for authors to carefully consider and respond to the feedback provided by reviewers to improve the quality of their work. Ignoring or failing to address the comments and concerns of the reviewers is a serious issue. The authors MUST describe what specific changes have been made in response to the points in the revised manuscript and on which pages; send us your revised paper with tracking changes too.

Overall, significant revisions are still needed in order to make this manuscript suitable for publication. The authors must provide a more thorough explanation of the novelty and significance of their work, and they must address the specific concerns raised by the referee in a clear and concise manner. Without these changes, it is unlikely that the manuscript can be able to meet the standards for publication in this journal. This may result in a rejection recommendation.

Author Response

I have already revised the manuscript and previous report. Thank you very much again for your comments and suggestions. Please see the response in the attachment.
